# Non-Volatile Reconfigurable Compact Photonic Logic Gates Based on Phase-Change Materials

**DOI:** 10.3390/nano13081375

**Published:** 2023-04-15

**Authors:** Yuqing Zhang, Zheng Peng, Zhicheng Wang, Yilu Wu, Yuqi Hu, Jiagui Wu, Junbo Yang

**Affiliations:** 1College of Artificial Intelligence, Southwest University, Chongqing 400715, China; zyq0419@email.swu.edu.cn (Y.Z.); pengzheng97@email.swu.edu.cn (Z.P.); zcwang1027@foxmail.com (Z.W.); imwuyilu@163.com (Y.W.); hyq895651004@163.com (Y.H.); 2Center of Material Science, National University of Defense Technology, Changsha 410073, China; 3School of Physical Science and Technology, Southwest University, Chongqing 400715, China

**Keywords:** photonic logic gates, phase-change material, non-volatility, reconfigurability

## Abstract

Photonic logic gates have important applications in fast data processing and optical communication. This study aims to design a series of ultra-compact non-volatile and reprogrammable photonic logic gates based on the Sb_2_Se_3_ phase-change material. A direct binary search algorithm was adopted for the design, and four types of photonic logic gates (OR, NOT, AND, and XOR) are created using silicon-on-insulator technology. The proposed structures had very small sizes of 2.4 μm × 2.4 μm. Three-dimensional finite-difference time-domain simulation results show that, in the C-band near 1550 nm, the OR, NOT, AND, and XOR gates exhibit good logical contrast of 7.64, 6.1, 3.3, and 18.92 dB, respectively. This series of photonic logic gates can be applied in optoelectronic fusion chip solutions and 6G communication systems.

## 1. Introduction

All-optical networks, optical computing, optical switches, and optical transmission systems comprise core units based on photonic logic gates, allowing the realization of all-optical signal processing [1,2,3]. Simultaneously, as artificial intelligence, big data, and other technologies advance, all-optical logic gates provide new methods for the creation of new kinds of computers. Therefore, photonic logic gates have recently been extensively examined, and various methods, including photonic crystals [4,5,6], semiconductor amplifiers [7,8], and implementation based on multimode interference, have been suggested for their realization [9,10,11,12]. Although these schemes offer greater contrast ratios (CR), they still struggle to achieve ultra-small size and high integrability, and their functions are preset and cannot be reprogrammed.

To meet the demand for reconfigurable and easily integrable devices, optical phase-change materials (O-PCMs) are promising candidates that offer important benefits such as reconfigurability, low loss, non-volatility, and a significant difference in complex refractive index (*ñ* = *n* + *i**k*) between the amorphous and crystalline states [13,14]. Pump light excitation or electrical heating may be used to regulate the phase transition [15,16,17], and the transition between the two states is a multi-stage phase transition process that possesses non-volatile characteristics. Moreover, one of the most essential characteristics of O-PCMs is their refractive index advantage. O-PCMs exhibit variations in refractive index in response to changes in temperature or other external variables, allowing for the fabrication of optical devices that can adapt to such changes. The rapid rate of change in their refractive indices, with response times in the order of milliseconds or even microseconds [18], offers additional advantages for the construction of faster and more flexible optical systems. As a result, O-PCMs are a promising avenue for improving the efficiency and adaptability of optical systems. O-PCMs, such as Ge_2_Sb_2_Te_5_ (GST) [16,19], Ge_2_S_2_Se_4_Te_1_ (GSST) [20], and Sb_2_Se_3_ [21,22], are widely used in many optical devices, including optical switches [20,23,24,25], optical memories [26,27], mode converters [28], programmable metasurfaces [29,30], and power beam splitters [31]. The GST absorption loss in the crystalline form (*k* = 1.49) and in the amorphous state (*k* = 0.12) is quite substantial, resulting in an excessive loss at 1550 nm; however, there is a noticeable difference in the refractive index between the two states. The Te component is substituted with Se in GSST. Despite improvements, it still exhibits a high absorption loss (*k* = 18 × 10^−4^ in the crystalline state and *k* = 0.35 in the amorphous form). In contrast, Sb_2_Se_3_ has the advantages of high refractive index contrast between the crystalline and amorphous states (∆n ≈ 0.77) [18,32], low loss at 1550 nm (absorption loss *k* ≈ 0 for both crystalline and amorphous states), and high integrability, which make it an ideal O-PCM for telecommunication band programmable optics [22,33]. Additionally, since the refractive index of Sb_2_Se_3_ is close to that of silicon, it can be integrated into standard silicon-on-insulator (SOI) integrated photonic platforms [22]. For instance, Matthew Delaney et al. constructed programmable silicon photonic circuits in 2021 utilizing PCM Sb_2_Se_3_ patch deposition over a 220 nm thick SOI waveguide.

To increase the design efficiency of optical devices and alleviate the problem of integration owing to their large size, inverse design techniques such as the direct binary search (DBS) algorithm [15,34,35], genetic algorithm [36], adjoint method [20,37], and objective-first design algorithm [38] have become increasingly important research topics. The inverse design approach can explore a bigger parameter space for higher-performance and more compact photonic devices than the conventional forward design method. Inverse design mainly adjusts the amplitude, phase, and polarization of the light field by changing the local refractive index of the structure [39] and achieves overall control by manipulating the optical characteristics of numerous unit pixels to efficiently realize various required functional devices. The DBS, as an inverse design algorithm, is widely used in the design of various nanophotonic devices, such as power dividers [40,41], mode multiplexers [42], and wavelength demultiplexers [43].

In this study, based on the above theory, we aimed to design and simulate an Sb_2_Se_3_-based ultra-compact photonic logic gate with nonvolatile and reconfigurable characteristics. Section 2 presents the material and structure of the designed photonic logic gate. Section 3 presents the results of numerical simulations of the adopted design. Finally, Section 4 concludes the paper.

## 2. Theory and Design of Photonic Logic Gate

Figure 1a shows a graphical schematic of the logic gate. To create the circumstance where waveguide A and B have no light source input while there exists light output in the output waveguide, a control waveguide C with a constant light source input is set up during the design process. Figure 1b shows the logic gate structure. As shown in Figure 1c, two structures, I and II, are designed by DBS, and the design scheme in Figure 1b is adopted for both structures. Here, 220 nm thick Si and 3 μm thick SiO_2_ layers are used as a standard silicon-on-insulator. The structure has three input and one output waveguides with a width of 400 nm, spacing of 400 nm between the input waveguides, and intermediate design area of 2.4 μm × 2.4 μm, which is divided into 24 × 24 pixels, each with a size of 100 nm × 100 nm. Each pixel state is Si or Sb_2_Se_3_, where the Sb_2_Se_3_ and Si materials are represented by yellow and blue pixel cells, respectively. By switching between the crystalline and amorphous phases of Sb_2_Se_3_, different logic gates can be created. The device with structure I is a NOT gate when Sb_2_Se_3_ is in the crystalline state, and it is an OR gate when Sb_2_Se_3_ is in the amorphous state. The device with structure II is an XOR gate when Sb_2_Se_3_ is in the crystalline state, and it is an AND gate when Sb_2_Se_3_ is in the amorphous state. The extinction coefficient and refractive index curves of Sb_2_Se_3_ in the C-band are shown in Figure 1d, which demonstrates the exceptionally low loss of Sb_2_Se_3_ at 1550 nm in both the crystalline and amorphous phases, as well as the stark contrast in the refractive indices of the two states.

The DBS algorithm’s optimization process is shown in Figure 2. The figure of merit (FOM) of the logic gate was initially calculated using a randomized beginning structure, starting from the first pixel. This cell state was then switched, and the FOM of the logic gate was calculated again. If the FOM was improved, the state of the cell was retained. If the FOM was not improved, the cell was restored to its original state. We then determined the subsequent cells. Each cell was calculated individually until the performance of the existing structure satisfied the required criteria.

In addition, the performance of photonic logic gates can be measured by calculating their CR [5]. Under the same conditions, the higher the CR, the better the performance of the device. The CR is defined as:(1)$$CR\left(dB\right)=10log(\frac{P_{ON}}{P_{OFF}})$$
where P_ON_ represents the minimum output power when the logic state of the logic gate is “1”, and P_OFF_ represents the maximum output power when the logic state of the logic gate is “0” [44].

## 3. Simulation Results

### 3.1. Photonic OR Logic Gate

Figure 3 shows the design of the photonic OR logic gate for structure I in the amorphous state of Sb_2_Se_3_. A graphical depiction of the gate is shown in Figure 3a, and its structural distribution is shown in Figure 3b. The OR logic gate was designed with two input ports, one output port, and a control port C, which maintains a continuous input optical signal. In the simulation, we use pulse light as the input signal and an SM waveguide as the waveguide for the input signal. Figure 3c–f show the optical field intensity distributions of the four logical operation states of the OR logic gate at a 1550 nm wavelength, and the contrast between logic “1” and logic “0” can be seen. For the design, we adoptd a 2.4 μm × 2.4 μm area and specified the logic operation as the objective function to be optimized using the DBS algorithm. The power transfer mechanism of the logic gate was created by modulating the refractive index inside the optical logic gate, leading to the reflection or refraction of light and thus enabling the realization of an optical field effect, as illustrated in Figure 3c–f. Figure 3g displays the OR logic gate output power at a working wavelength of 1550 nm. The designed logic gate can be seen to have excellent stability across the entire C-band. Table 1 lists information on the binary output truth table and the optical power output of the OR logic gates. Each input light source in the simulation has the same power, which is P_in_. We set the threshold value for logic “1” and “0” at 0.5 P_in_. When the output optical power is less than 0.5 P_in_, the logic state is expressed as logic “0”, and when it is greater than 0.5 P_in_, the logic state is expressed as logic “1”. When the logic “1” of the output power is greater than the threshold and the logic “0” is less than the threshold, even if there is a reading error, the device’s function can still be achieved. When the input signal of the two input waveguides is “0”, the output optical power is 0.13 P_in_, which is expressed as logic “0”. The output optical power is 0.78 P_in_, which is expressed as logic “1”, when the input signal of waveguide A is “0” and that of waveguide B is “1”. The output optical power is 0.82 P_in_, which is expressed as logic “1”, when the input signal of waveguide A is “1” and that of waveguide B is “0”. The output optical power is 2.05 P_in_, which is expressed as logic “1”, when the input signals of waveguides A and B are “1”. Calculated with the lowest logic “1” power and highest logic “0” power, the OR logic gate CR is 6.1 dB.

In Table 2, we compared the proposed OR logic gate with structures in the literature. Evidently, the proposed logic gate has a more compact structure and higher CR than other logic gates (including tunable logic gates) reported in the literature.

### 3.2. Photonic NOT Logic Gate

Figure 4 shows the design of a photonic NOT logic gate for structure I when Sb_2_Se_3_ is in the crystalline state. A graphical representation of the NOT logic gate is shown in Figure 4a. The distribution of the NOT logic gate structure is shown in Figure 4b. Port B serves as the input source port, Port C as the control port, and Port Y as the output port in the NOT gate. Furthermore, port C maintains a constant light source input. The light field intensity distribution of the NOT logic gate at 1550 nm is shown in Figure 4c,d. The logic state of output port Y displays logic “1” when there is no light source input to input port B. The state of output port Y displays logic “0” when a light source is input at input port B. The output power of the NOT logic gate is shown in Figure 4e for the entire C-band. Evidently, the output power is steady across the C-band.

The designed photonic NOT logic gate output optical power and truth table data are listed in Table 3. The threshold values for logic “1” and “0” are 0.5 P_in_, and the logic state is shown as logic “0” when the output optical power is less than 0.5 P_in_ and logic “1” when the output optical power is more than 0.5 P_in_. The CR is 6.1 dB. A comparison of the proposed NOT gate and some structures from the literature is presented in Table 4. The proposed structure is nonvolatile, compact, and has a high CR value.

### 3.3. Photonic AND Logic Gate

Figure 5 shows the design of a photonic AND logic gate for structure II when Sb_2_Se_3_ is in the amorphous state. A graphical depiction of the AND logic gate is shown in Figure 5a. The construction of the designed AND logic gate is illustrated in Figure 5b. The AND logic gate has two input ports (A and B), one output port (Y), and one control port (C). Figure 5c–f display the light field intensity distributions for all logic AND operations. The logic threshold is set at 0.6 P_in_; the output power is expressed as logic “1” when the output power is greater than 0.6 P_in_; and as logic “0” when the output power is less than 0.6 P_in_. As shown in Figure 5c, when there is no light source input to the input waveguide, the output power is 0.16 P_in_, which is expressed as logic “0”. As shown in Figure 5d, when waveguide A has no light source input and waveguide B has light source input, the output power is 0.41 P_in_, which is expressed as logic “0”. As shown in Figure 5e, when waveguide B has no light source input and waveguide A has light source input, the output power is 0.48 P_in_, which is expressed as logic “0”. As shown in Figure 5f, when the input waveguide has light source input, the output power is 1.03 P_in_, which is expressed as logic “1”. Evidently, the AND logic gate has a stable output power in the C-band, as shown in Figure 5g, which displays the output optical power of all its logic operations in the C-band. Table 5 provides specific information regarding the logical AND operations at a working wavelength of 1550 nm. This structure is contrasted in Table 6 with some structures from the literature. The proposed structure is more compact and reconfigurable.

### 3.4. Photonic XOR Logic Gate

A photonic XOR logic gate is created for structure II when Sb_2_Se_3_ is in the crystalline state, as shown in Figure 6. A graphical representation of the XOR logic gate is shown in Figure 6a. The structure of the XOR logic gate, depicted in Figure 6b, consists of a control waveguide (C), two input ports (A and B), and one output port (Y). The four logic operation states of the XOR logic gate are depicted in Figure 6c–f, which also illustrate the light field intensity distribution of the logic XOR operation at a wavelength of 1550 nm. When the output power is less than 0.3 P_in_, the logic state is expressed as logic “0”, and when the output power is more than 0.3 P_in_, the logic state is expressed as logic “1.” When the input waveguide does not have a light source input, as shown in Figure 6c, the output power is 0.004 P_in_, which is expressed as logic “0”. As shown in Figure 6d, the output power is 0.4 P_in_, which is expressed as logic “1,” when waveguide A has no light source input and waveguide B has a light source input. When waveguide A has a light source input but waveguide B does not, as illustrated in Figure 6e, the output optical power is 0.39 P_in_, which is expressed as the logic “1” state. When the input waveguide has a light source input, as illustrated in Figure 6f, the output optical power is 0.005 P_in_, which is expressed as logic “0”. The results of the photonic XOR gate logic operations are displayed in Figure 6e for the C-band. The XOR logic gate can clearly be seen to exhibit high stability and is not wavelength-sensitive in the C-band. The precise information and thresholds for the output power of each logical XOR operation at a wavelength of 1550 nm are listed in Table 7. A comparison of the proposed XOR logic gate and some structures from the literature is presented in Table 8.

Time propagation delay (TPD) and manufacturing tolerance analysis are considered essential criteria for evaluating the performance of logic gates, and both aspects have been simulated and discussed in the following section. 

TPD is a crucial parameter for evaluating logic gates, and we have determined the TPDs for the AND, NOT, XOR, and OR logic gates via simulation: 0.6 ps, 0.8 ps, 0.8 ps, and 0.6 ps, respectively. The results indicate that our device exhibits a response time of less than 1 ps, which meets the requirements for high-speed data processing.

Manufacturing tolerance refers to the minor deviations or alterations in the geometric shape of a structure that may occur during the manufacturing process due to factors such as materials and fabrication technology. To assess the effects of manufacturing tolerances on the structure, we conducted simulations of the light field of phase-change materials with varying side lengths ranging from −3 nm to +3 nm. The output optical power diagram of the all-optical logic gate under manufacturing tolerance is presented in Figure 7. Figure 7a–f show the output optical power diagram of the OR logic gate and the NOT logic gate in the C-band. It can be seen from the figure that disturbances in the size of the phase-change material has little impact on the output power of the structure. Additionally, we determined that the minimum contrast ratios for the OR and NOT logic gates at an operating wavelength of 1550 nm were 6.1 dB and 4.3 dB, respectively, under this manufacturing error. Overall, these results suggest that the structure we designed exhibits good manufacturing tolerances.

## 4. Conclusions

We designed non-volatile, reconfigurable, and ultra-compact photonic logic gates of size 2.4 μm × 2.4 μm using an inverse design approach and two states of phase-change material. The structure has four ports—two input ports, one output port, and one control port—and realizes four logic gates: OR, NOT, AND, and XOR. According to the results of the three-dimensional finite-difference time-domain simulations, photonic logic gates for OR, NOT, AND, and XOR are realized in a wavelength range of 35 nm from 1550 nm, with CR values of 7.64, 6.1, 3.3, and 18.92 dB, respectively. The ultra-small size of the structures we developed has resulted in a TPD of less than 1 ps, which satisfies the requirements for high-speed data processing. Moreover, we compared our logic gate structures with those presented in other literature and found that several of their benefits, including reconfigurability, ultra-compactness, and high CR, are shared by our structures. We also conducted a manufacturing tolerance study, which indicated that our structure exhibits excellent manufacturing tolerances. It is crucial for the development of photonic integrated circuits and photonic signal-processing nanocircuits that this series of architectures combine the advantages of reconfigurability, ultra-compactness, and high CR. 

## Figures and Tables

**Figure 1 nanomaterials-13-01375-f001:**
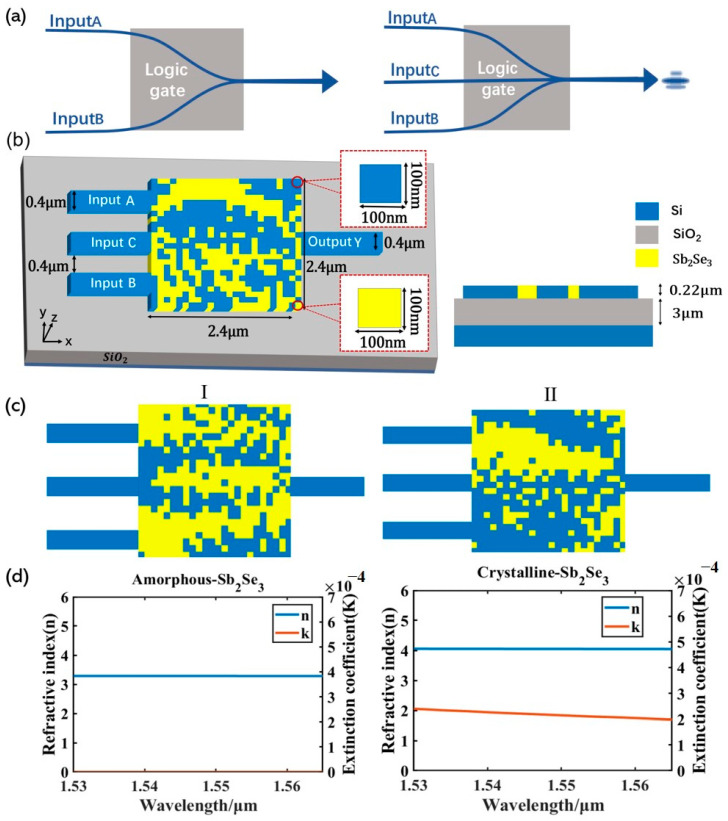
Photonic logic gates design. (**a**) Graphical representation of the photonic logic gate; (**b**) schematic diagram of structural parameters, a C-band transverse wave (TE mode) is coupled from the left input waveguides to the right output waveguide through the optimized region; (**c**) schematic diagram of structures I and II; (**d**) Sb_2_Se_3_’s refractive index and extinction coefficient for wavelengths between 1530 and 1565 nm in both its crystalline and amorphous phases.

**Figure 2 nanomaterials-13-01375-f002:**
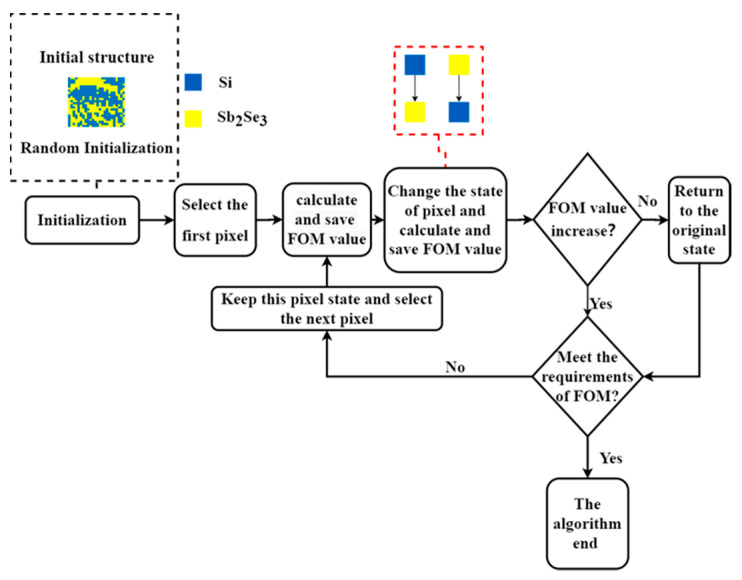
DBS algorithm flowchart.

**Figure 3 nanomaterials-13-01375-f003:**
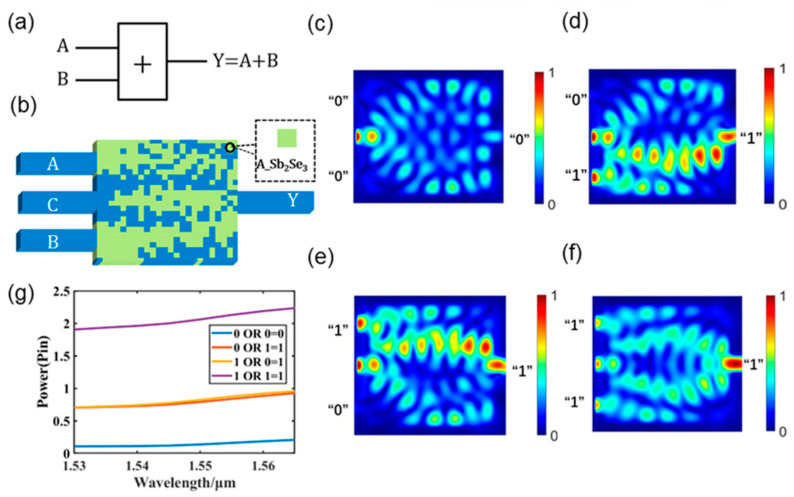
Photonic OR logic gate. (**a**) Graphical symbols of OR logic gate; (**b**) designed OR logic gate; (**c**–**f**) light field intensity distributions of all operations of the OR logic gate; (**g**) output optical power diagram of the OR logic gate in the C-band.

**Figure 4 nanomaterials-13-01375-f004:**
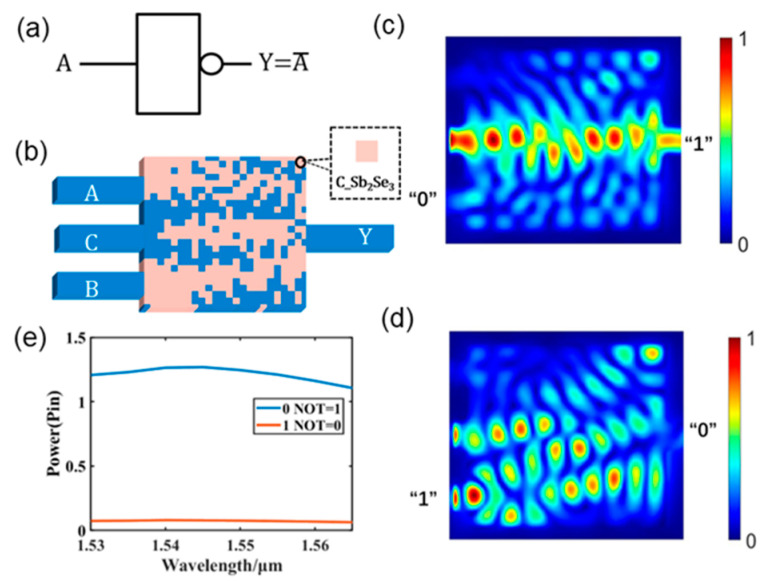
Photonic NOT logic gate. (**a**) Graphical symbols of the NOT logic gate; (**b**) the designed NOT logic gate; (**c**,**d**) light field intensity distributions in all operations of the NOT logic gate; (**e**) output optical power diagram of the NOT logic gate in the C-band.

**Figure 5 nanomaterials-13-01375-f005:**
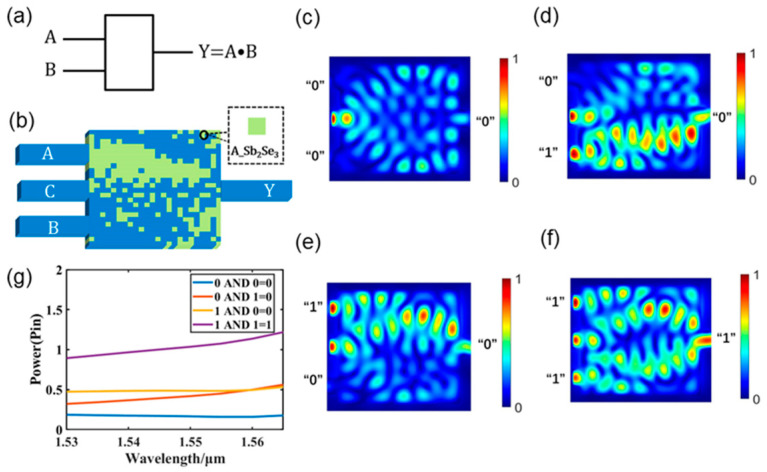
Photonic AND logic gate. (**a**) Graphical symbols of the AND logic gate; (**b**) designed AND logic gate; (**c**–**f**) light field intensity distributions of all operations of the AND logic gate; (**g**) output optical power diagram of the AND logic gate in the C-band.

**Figure 6 nanomaterials-13-01375-f006:**
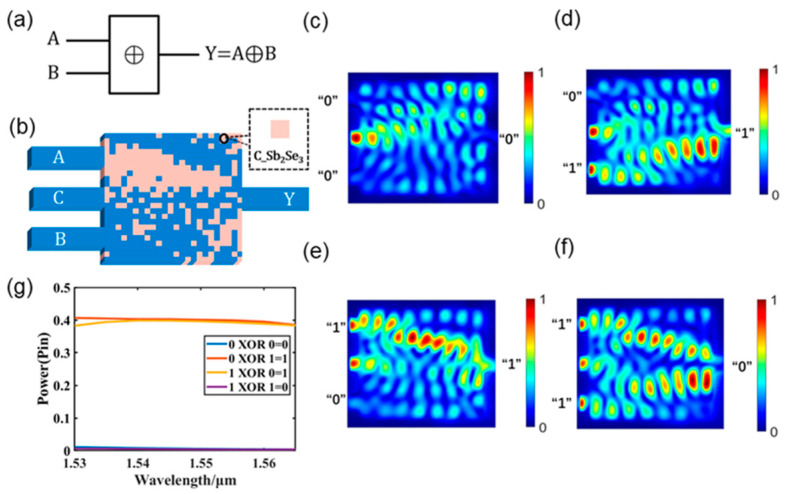
Photonic XOR logic gate. (**a**) Graphical symbols of the XOR logic gate; (**b**) designed XOR logic gate; (**c**–**f**) light field intensity distributions of all operations of the XOR logic gate; (**g**) output optical power diagram of the XOR logic gate in the C-band.

**Figure 7 nanomaterials-13-01375-f007:**
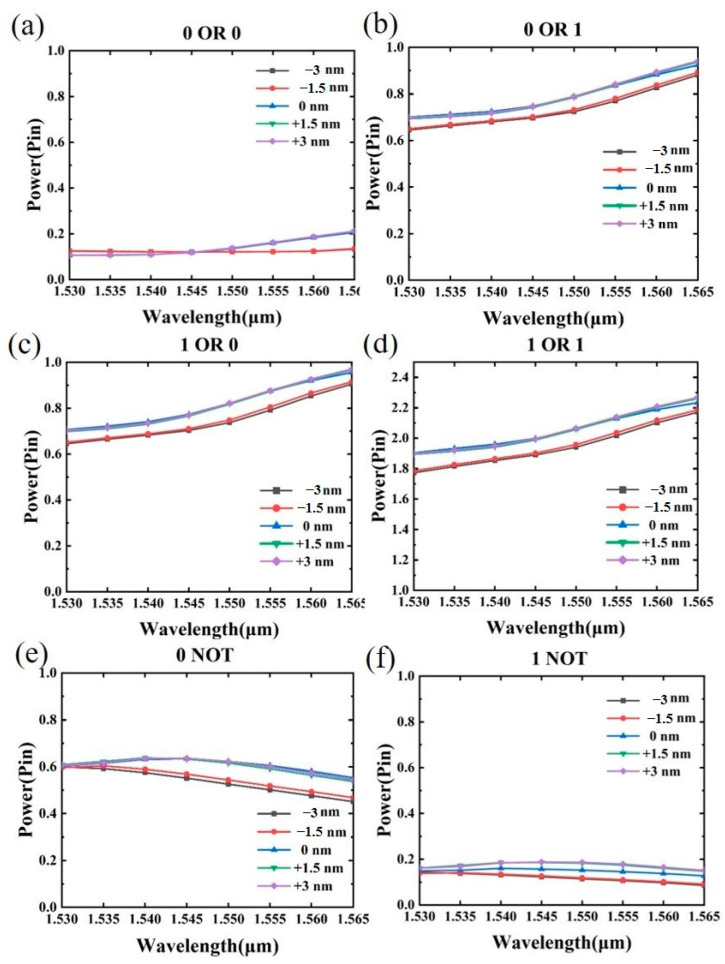
Analysis of manufacturing tolerances for all-optical logic gates. (**a**–**f**) A graph of the output optical power of OR logic gates and NOT logic gates when the phase-change material edge length varies from −3 nm to +3 nm.

**Table 1 nanomaterials-13-01375-t001:** Photonic OR logic gate power and binary output truth table data.

OR Gate: Y = A + B.					
Input A	Input B	Input C	Threshold	Output Power of Y	Binary Output
0	0	*P_in_*	0.5 *P_in_*	0.13 *P_in_*	0
0	*P_in_*	*P_in_*	0.5 *P_in_*	0.78 *P_in_*	1
*P_in_*	0	*P_in_*	0.5 *P_in_*	0.82 *P_in_*	1
*P_in_*	*P_in_*	*P_in_*	0.5 *P_in_*	2.05 *P_in_*	1

**Table 2 nanomaterials-13-01375-t002:** Performance comparison between the proposed photonic OR logic gate and several reference structures reported in other papers.

Photonic OR Logic Gate					
Ref.	Operating Wavelength (nm)	Size (μm^2^)	CR (dB)	Function Switchable	Non-Volatile
[45]	1300	5.016	5.78	No	No
[46]	1550	0.16	12.55	Yes	No
[47]	1550	132.6	6.02	No	No
This work	1550	5.76	7.64	Yes	Yes

**Table 3 nanomaterials-13-01375-t003:** Photonic NOT logic gate power and binary output truth table data.

NOT Gate: Y = $\bar{B}$					
Input A	Input B	Input C	Threshold	Output Power of Y	Binary Output
/	0	*P_in_*	0.5 *P_in_*	0.62 *P_in_*	1
/	*P_in_*	*P_in_*	0.5 *P_in_*	0.15 *P_in_*	0

**Table 4 nanomaterials-13-01375-t004:** Performance comparison between the proposed photonic NOT logic gate and several reference structures reported in other papers.

Photonic NOT Logic Gate					
Ref.	Operating Wavelength (nm)	Size (μm^2^)	CR (dB)	Function Switchable	Non-Volatile
[48]	1550	122	5	No	No
[45]	1300	5.016	1.19	No	No
[49]	1550	558	20	No	No
This work	1550	5.76	6.1	Yes	Yes

**Table 5 nanomaterials-13-01375-t005:** Photonic AND logic gate power and binary output truth table data.

AND Gate: Y = A • B					
Input A	Input B	Input C	Threshold	Output Power of Y	Binary Output
0	0	*P_in_*	0.5 *P_in_*	0.16 *P_in_*	0
0	*P_in_*	*P_in_*	0.5 *P_in_*	0.41 *P_in_*	0
*P_in_*	0	*P_in_*	0.5 *P_in_*	0.48 *P_in_*	0
*P_in_*	*P_in_*	*P_in_*	0.5 *P_in_*	1.03 *P_in_*	1

**Table 6 nanomaterials-13-01375-t006:** Performance comparison between the proposed photonic AND logic gate and several reference structures reported in other papers.

Photonic AND Logic Gate					
Ref.	Operating Wavelength (nm)	Size (μm^2^)	CR (dB)	Function Switchable	Non-Volatile
[45]	1300	5.016	4.7	No	No
[50]	1550	168	8.45	No	No
[51]	1550	110	6.9	No	No
[47]	1550	122.96	6.02	No	No
This work	1550	5.76	3.31	Yes	Yes

**Table 7 nanomaterials-13-01375-t007:** Photonic XOR logic gate power and binary output truth table data.

XOR Gate: Y = A ⊕ B					
Input A	Input B	Input C	Threshold	Output Power of Y	Binary Output
0	0	*P_in_*	0.3 *P_in_*	0.005 *P_in_*	0
0	*P_in_*	*P_in_*	0.3 *P_in_*	0.4 *P_in_*	1
*P_in_*	0	*P_in_*	0.3 *P_in_*	0.39 *P_in_*	1
*P_in_*	*P_in_*	*P_in_*	0.3 *P_in_*	0.004 *P_in_*	0

**Table 8 nanomaterials-13-01375-t008:** Performance comparison between the proposed photonic XOR logic gate and several reference structures reported in other papers.

Photonic XOR Logic Gate					
Ref.	Operating Wavelength (nm)	Size (μm^2^)	CR (dB)	Function Switchable	Non-Volatile
[4]	1550	252	19.95	Yes	No
[45]	1300	5.016	1.76	No	No
[52]	—	729	9.33	Yes	No
[53]	1550	265	5.67	Yes	No
This work	1550	5.76	18.92	Yes	Yes

## Data Availability

Data available in a publicly accessible repository.

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
