# Peer review of "Non-Volatile Reconfigurable Compact Photonic Logic Gates Based on Phase-Change Materials"

_nanomaterials, 2023, doi:10.3390/nano13081375_

Round 1
Reviewer 2 Report
In their work, Zhang and coauthors design photonic logic gates based on Sb2Se3 phase change material. Overall the manuscript seems technically correct, hence would deserve publication, but I feel the introduction and conclusion, as well as the comparison with other works in the literature:
- I agree that the material properties are promising for the purpose of photonic integrated computing, however I feel that the discussion on the compatibility with SOI platform is lacking. In particular, technical challenges for the actual realisation of such devices should be discussed.
- I feel that the comparison with existing works might be unfair because comparing experimental results with conceptual propositions doesn´t take into account experimental nonidealities and fabrication defects. An added discussion on the robustness of the proposed design against such nonideality sources would improve the quality of the manuscript.
Reviewer 3 Report
In this paper, the authors introduced a Photonic logic gates have important applications in fast data processing and optical communication. This study aims to design a series of ultra-compact non-volatile and reprogrammable photonic logic gates based on the Sb2Se3 phase-change material. A direct binary search algorithm is adopted for the design, and four types of photonic logic gates (OR, NOT, AND, and XOR) are realized using the silicon-on-insulator technology.The idea behind this is interesting. However, I still have quite a number of concerns in this manuscript. There are times where there are not enough data to support the conclusions of the author. Please see some of the major concerns below.
1.The information for the Photonic logic gates design (figure 1) is not enough. The authors should give much more information about this. So the readers can get its reproducibility also the figure cannot be seen because of the small size. Thus, I recommend to improve the font size or the figure resolution. What is the input signal are this Gaussian beam CW or pulse ?
Are the source is the SM waveguide ?
2. The authors should give much more information about the novelty of this paper, especially the effect of using optical logic gates, which applications can be used this device?
3. The fabrication tolerance analysis, which can offer a good guide for the fabrication requirement, and the key parameters, need to be added in the results section.
4. What is the TPD for each gate ?
5. More references need to be included in the introduction part to understand the applications of using optical logic gates system devices and fibers for realizing an optical devices.
a. Combining Four Gaussian Lasers Using Silicon Nitride MMI Slot Waveguide Structure - Micromachines, 2022
b. A three demultiplexer C-band using angled multimode interference in GaN–SiO2 slot waveguide structures
- Nanomaterials, 2020
6. Much more discussion about the results should be given in this paper, especially the author needs to provide enough physicals mechanism analysis about the results.
Round 2
Reviewer 1 Report
the paper can be published as it is
Reviewer 3 Report
The new version can be published.